# MYC Targets Scores Are Associated with Cancer Aggressiveness and Poor Survival in ER-Positive Primary and Metastatic Breast Cancer

**DOI:** 10.3390/ijms21218127

**Published:** 2020-10-30

**Authors:** Amy Schulze, Masanori Oshi, Itaru Endo, Kazuaki Takabe

**Affiliations:** 1Breast Surgery, Department of Surgical Oncology, Roswell Park Comprehensive Cancer Center, Buffalo, New York, NY 14263, USA; amyschul@buffalo.edu (A.S.); Masanori.Oshi@RoswellPark.org (M.O.); 2Department of Surgery, Jacobs School of Medicine and Biomedical Sciences, State University of New York, Buffalo, New York, NY 14263, USA; 3Department of Gastroenterological Surgery, Yokohama City University Graduate School of Medicine, Yokohama 236-0004, Japan; endoit@yokohama-cu.ac.jp; 4Department of Surgery, Niigata University Graduate School of Medical and Dental Sciences, Niigata 951-8510, Japan; 5Department of Breast Surgery and Oncology, Tokyo Medical University, Tokyo 160-8402, Japan; 6Department of Gastrointestinal Tract Surgery, Fukushima Medical University School of Medicine, Fukushima 960-1295, Japan

**Keywords:** breast cancer, biomarker, MYC, tumor gene expression

## Abstract

*MYC* is one of the most studied oncogenes that is known to promote cell proliferation. We utilized MYC targets v1 and MYC targets v2 scores of gene set variation analysis and hypothesized that these scores correlate with tumor aggressiveness and survival outcomes. We examined a total of 3109 breast cancer patients from TCGA, METABRIC, and GSE124647 cohorts. In each cohort, the patients were divided into high- and low-score groups using the upper third value as the cut off. As expected, higher scores were related to increased cell proliferation and worse clinical and pathologic features. High MYC targets scores were associated with worse survival, specifically in primary ER-positive breast cancer, consistently in both TCGA and METABRIC cohorts. In ER-positive breast cancer, high MYC targets v1, but not v2 score, was associated with high mutation load, and high MYC targets v1 and v2 scores were both associated with increased infiltration of pro- and anti-cancerous immune cells. We found that high MYC scores were associated with worse survival in metastatic breast cancer. Our findings show that the MYC targets v1 and v2 scores are associated with tumor aggressiveness and poor prognosis in ER-positive primary tumors, as well as in metastatic breast cancer.

## 1. Introduction

Breast cancer is currently the most prevalent cancer among women in the United States and it is estimated that 40,000 women in the US will die from the disease in 2020 [1]. Breast cancer is categorized into three subtypes—estrogen receptor (ER)-positive, human epidermal receptor 2 (HER2)-positive and triple-negative breast cancer (TNBC). Of the subtypes, ER-positive cancer is the most common and, although it is associated with the best prognosis of the three subtypes, there remains a significant long-term recurrence rate of 10–41% depending on tumor pathological grade, tumor size, and lymph node metastasis status [2]. It has been shown that even with breast cancers of similar stage and subtype, there is variability in clinical response to treatment, opening the possibility of personalized treatment of a patient’s specific breast cancer [3]. Knowing this, it is important to have a prognostic biomarker which can identify the tumors that are more likely to be aggressive in order to potentially tailor treatment for those patients.

There are currently two multigene risk scores that are routinely used in patient care, Oncotype and MammaPrint. Although both have been shown to have prognostic use in the clinical setting, there remain limitations. Mainly, the genes studied are of a wide variety and the number of genes studied remains 16 for oncotype and 70 for MammaPrint [3,4], thus the results do not elaborate therapeutic targets. The technological advances in recent years for genetic analyses of tumors, as well as the subsequent cost reduction, creates the opportunity for use of a prognostic score that is both more expansive in the number of genes evaluated and also more specific to one of the cancer hallmarks.

It is well known that there are multiple hallmarks of cancer and promoting cell proliferation is one of them [5]. MYC is a transcription factor that is known to regulate multiple human genes that promote cell growth and proliferation [6]. It can also alter apoptosis via alteration of the pro- and anti-apoptotic members of the BCL-2 family as well as activation of telomerase and it controls expression levels of vascular endothelial growth factor (VEGF) and thereby controls angiogenesis [7]. These downstream targets make *MYC* one of the most powerful oncogenes. Specifically in breast cancer, *MYC* overexpression has been well reported [8,9,10,11].

Our group has performed multiple studies in the past demonstrating the clinical relevance of prognostic scores for G2M checkpoint, E2F signaling, and CD8 [12,13,14]. Knowing the *MYC* oncogene has been identified previously as a source for uncontrolled cell proliferation, we used The Cancer Genome Atlas (TCGA) breast invasive carcinoma cohort [15] as well as the Molecular Taxonomy of Breast Cancer International Cohort (METABRIC) [16] and GSE124647 [17] cohorts in order to study the expression of hallmark MYC targets v1 and MYC targets v2 gene sets [18] and their correlation with tumor aggressiveness as well as immune cell infiltration and survival in metastatic disease. We hypothesized that higher scores of MYC targets v1 and MYC targets v2 would be associated with higher rates of cell proliferation, causing increased tumor aggressiveness and worse survival.

## 2. Results

### 2.1. High MYC Targets v1 and MYC Targets v2 Scores Correlate with Cell Proliferation

MYC targets v1 and MYC targets v2 scores were determined by gene set variation analysis (GSVA) of Molecular Signatures Database Hallmark gene sets, “HALLMARK_MYC_TARGETS_V1” and “HALLMARK_MYC_TARGETS_V2”, which are 2 of 50 gene sets initially created by Liberzon et al. [18] (see Appendix A). We used a similar method as used in our previous studies on KRAS signaling [19], the G2M cell cycle pathway [12], the E2F pathway [13], and angiogenesis [20]. The upper third value was used as a cut off to divide the tumors into high- or low-MYC score groups in each cohort. A table depicting the characteristics of each cohort by high and low MYC scores is shown in Appendix A.

Given that *MYC* is one of the most studied oncogenes, we expected that both MYC scores would correlate with cell proliferation, which was assessed by the proliferation score, precalculated in the TCGA cohort by Thorsson et al. [21], and expression of *MKI67* that encodes KI67, the most commonly used marker for cell proliferation in the clinic [22]. Indeed, the MYC targets v1 and v2 scores significantly correlated with the proliferation score in the TCGA cohort (MYC targets v1; *r* = 0.634, *p* < 0.01, MYC targets v2; *r* = 0.531, *p* < 0.01, respectively, Figure 1A). There was also significant but weaker correlation of both MYC v1 and v2 scores with *MKI67* expression (MYC targets v1; *r* = 0.45, *p* < 0.01, MYC targets v2; *r* = 0.366, *p* < 0.01, respectively, Figure 1A). High MYC v1 and v2 scores were associated with significantly high *MKI67* expression (both *p* < 0.001, Figure 1B). This was validated in the METABRIC cohort (both *p* < 0.001). These findings suggest that breast cancer with high MYC targets v1 or MYC targets v2 scores are associated with higher proliferation compared to low scores. Additionally, individual genes associated with the MYC target v1 and v2 scores were assessed for their correlation with proliferation and these values are shown in Appendix A. Correlation analysis was also performed for each gene and MYC score to assess the contribution of each gene to the overall outcome seen in the MYC target scores dataset and the Spearman *r* value is also shown in Appendix A. The *MYC* gene was individually evaluated in regards to correlation with proliferation and *MKI67* expression and was found to have significant correlations, although very weak compared with MYC targets v1 or v2 (*r* = 0.02, *p* < 0.01; *r* = 0.155, *p* < 0.01, respectively; Appendix A).

### 2.2. MYC Targets v1 and MYC Targets v2 Scores Are Elevated in Clinically Aggressive Breast Cancer

Given that high MYC scores were associated with high cell proliferation, it was expected that higher scores would also be associated with clinically aggressive breast cancer. Indeed, in the TCGA cohort, MYC targets v1 and v2 scores were both elevated in higher Nottingham pathological grade (MYC v1 and v2, both *p* < 0.001, Figure 2). High MYC targets scores were also associated with clinically aggressive subtypes, triple-negative breast cancer (TNBC) and HER2-positive breast cancer, compared with ER-positive/HER2-negative tumors (MYC v1 and v2, both *p* < 0.001). These results were verified in the METABRIC cohort, where both MYC v1 and v2 scores were elevated in higher American Joint Committee on Cancer (AJCC) cancer stage, higher grade, and TNBC or HER2-positive cancer (all *p* < 0.001).

### 2.3. High MYC Targets v1 and v2 Scores of Primary Breast Cancer Are Both Related to Worse Survival

Given that higher MYC scores for both v1 and v2 were related to worse clinical and pathologic features, we expected there would also be an association between high MYC scores of the primary breast cancer and worse survival. In the ER-positive/HER2-negative subtype of METABRIC cohort, high MYC targets v1 and MYC targets v2 scores were significantly associated with worse disease-specific survival (DSS) as well as disease-free survival (DFS) (MYC targets v1; *p* = 0.018, *p* = 0.048, MYC targets v2; *p* < 0.001, *p* < 0.001, respectively). High MYC targets v1 score, but not MYC targets v2, was associated with worse DSS in the TCGA cohort (MYC targets v1; *p* = 0.037, MYC targets v2; *p* = 0.078, respectively). There was no survival difference among either MYC targets v1 or v2 high and low groups in subtypes of TNBC or HER2-positive cancer (Figure 3). This suggests that MYC target scores are clinically relevant specifically in the ER-positive/HER2-negative subtype. Individual genes comprising the MYC scores were also analyzed using COX analysis for survival benefit to further examine each gene’s contribution to the overall score and these are listed in Appendix A.

### 2.4. High MYC Targets v1, But Not v2 Score, Is Associated with High Mutation Load in ER-Positive/HER2-Negative Breast Cancer

Given that high MYC targets v1 and v2 scores were associated with worse survival, specifically in the ER-positive/HER2-negative subtype, we investigated the relationship between MYC score in this subtype and mutation load assessed as copy number alteration (CNA), silent and nonsilent mutations, single-nucleotide variation (SNV), and insertion and deletion (Indel) neoantigen-utilizing dataset obtained from Thorsson et al. in the TCGA cohort [21] (Figure 4). There was significant association between high MYC targets v1 score and CNA (*p* < 0.001), silent mutation (*p* = 0.041), and SNV (*p* = 0.043). In the MYC targets v2 score, there was significant association between the high score and CNA alone (*p* < 0.001). These results suggest that both MYC targets v1 and v2 score are associated with copy number alteration, which may be because MYC target genes are activated when there is MYC amplification. In addition, because the *MYC* gene itself is known to impact mutation in part due to ability to induce reactive oxygen species (ROS) dependent on p53 status in vitro [23,24,25], we further analyzed the specific correlation between high *MYC* expression and ROS in both mutant and wildtype p53 groups, as well silent or nonsilent mutations but did not find significant difference (Appendix A).

### 2.5. High MYC Targets v1 and v2 Scores Are Associated with Increased Immune Cell Infiltration in ER-Positive/HER2-Negative Tumors

Knowing that high MYC scores were associated with high mutation scores in ER-positive/HER2-negative tumors, we predicted that there would also be increased immune cell infiltration in these tumors. We examined the immune cell composition of the tumors using xCell algorithm, and found that tumors with high MYC scores were significantly infiltrated with both pro-cancerous immune cells (regulatory T cells [Treg] and T helper type 2 cells [Th2]) as well as anti-cancerous immune cells (CD4+ T cells, CD8+ T cells, T helper type 1 cells [Th1], M1 macrophages, and dendritic cells [DC]). There were similar results seen in both the TCGA and METABRIC cohorts (Figure 5A).

We further examined the correlation of MYC targets v1 and v2 scores with abundance of Th1 or Th2 cells in the ER-positive/HER2-negative subtype (Figure 5B). In the TCGA cohort, there was a strong correlation of MYC targets v1 and v2 scores with Th1 cells (MYC v1; *r* = 0.745, *p* < 0.01, MYC v2; *r* = 0.811, *p* < 0.01, respectively). The correlation was relatively weaker with Th2 cells (MYC v1; *r* = 0.505, *p* < 0.01, MYC v2; *r* = 0.351, *p* < 0.01). These results were roughly validated by similar results in the METABRIC cohort, but with weaker correlation; Th1 cells (MYC v1; *r* = 0.053, *p* = 0.05, MYC v2; *r* = 0.492, *p* < 0.01, respectively), and Th2 cells (MYC v1; *r* = 0.62, *p* < 0.01, MYC v2; *r* = 0.358, *p* < 0.01, respectively).

### 2.6. High MYC Scores Are Associated with Worse Survival in Metastatic Breast Cancer

Given the association of disease-free survival and MYC scores of primary breast cancer, we predicted that higher MYC scores would also be associated with worse survival in metastatic breast cancer. The GSE124647 cohort was analyzed for progression-free survival (PFS) data as well as data on metastasis site (Figure 6). Indeed, high MYC targets v1 and v2 scores were both significantly associated with worse PFS in the whole cohort of metastatic tumors (MYC targets v1 *p* = 0.015, MYC targets v2 *p* = 0.011). However, there was no significant survival difference by score in specific metastatic sites except for MYC targets v2 score in liver metastasis (*p* < 0.001).

## 3. Discussion

For this study, we examined tumors in three large breast cancer patient cohorts (TCGA, METABRIC and GSE124647) and their MYC targets v1 and MYC targets v2 scores and the correlation with cell proliferation, clinical and pathologic features, survival differences, and immune cell infiltration. The MYC targets v1 and v2 scores were defined as the GSVA score of the “HALLMARK_MYC_TARGETS_V1” and “HALLMARK_MYC_TARGETS_V2” gene sets using the upper third value as the cut off [18]. High MYC targets v1 and v2 scores were associated with higher cell proliferation and correlated with higher expression of *MKI67*, which was verified in both TCGA and METABRIC cohorts. Higher MYC scores were also associated with higher stage, pathological grade, and worse subtype. We found that higher MYC scores were associated with worse survival, but only in ER-positive/HER2-negative tumors, as there was no survival difference in TNBC or HER2-positive tumors. Specifically looking at ER-positive tumors, we found that there was a higher level of mutations associated with high MYC scores. We then examined the tumor immune microenvironment and found that there was significant infiltration of both favorable and unfavorable immune cells in higher MYC score tumors as well as correlation of higher T helper cells. There was a significant correlation between high MYC v1 and v2 scores and worse progression-free survival in metastatic breast cancer.

The hallmark gene sets were initially published by Liberzon et al. in 2015 [18]. These gene sets were created from a hybrid approach using both manual review and computational processing which allows for minimal redundancy while still correlating with corresponding phenotypes which increases their biological relevance. The two hallmark gene sets MYC_TARGETS_V1 and MYC_TARGETS_V2 were, like the other hallmark sets, created from multiple founder sets. However, these sets are distinctly separate in the initial study and have separate datasets associated with them. Since the time of initial publication, the article has been cited over 1200 times. We utilized MYC targets v1 and v2 gene sets as the scores since they are well accepted and authenticated gene sets.

Unregulated cell proliferation is one of the most studied components of the hallmarks of cancer [5]. Given the *MYC* oncogene is known to control multiple aspects of cell cycle regulation [26], we were interested in the clinical significance of elevated levels of *MYC* specifically in breast cancer. Current genomic tests that are available include Oncotype DX and MammaPrint. Both of these tests examine levels of gene expression and scores the cancer into high or low risk of recurrence in order to aid decision making on whether or not the patient will benefit from adjuvant chemotherapy [27,28]. Given that a large amount of ER-positive patients receive unnecessary chemotherapy [28], predictive biomarkers such as these are extremely useful in tailoring treatment based on molecular aspects of a patient’s tumor in addition to clinical and pathologic characteristics. However, there are still limitations of these tests. Specifically, of the genes studied in the Oncotype, there are only 5 related to cell proliferation and even of the 70 genes examined in the MammaPrint, only 12 are specifically related to cell proliferation. The MYC targets scores assess 200 genes in v1 and another 58 in v2. This analysis of a larger number of genes specifically related to cell proliferation may be more useful at assessing tumor aggressiveness. In addition, patient populations that may benefit from the Oncotype or MammaPrint are limited. For example, Oncotype does not identify which chemotherapy the patient will benefit from most. Similarly, MammaPrint has only been approved for use in node-negative, stage I or II disease [29]. With further study, the MYC targets scores is expected to be useful in patients with accelerated cell proliferation, which allow more specific patient selection.

The tumor immune microenvironment has been shown to play a role in tumor aggression as well as in predicting recurrence in breast cancer [30,31,32]. It is known that specifically in ER-positive breast cancers, late recurrence is particularly pronounced when compared to other subtypes [30,33]. In this study, the increased amount of both favorable and unfavorable immune cells likely relates to the worse survival that was observed in metastatic disease in the high MYC groups.

The ER-positive/HER2-negative subtype of breast cancer is known to be less susceptible to chemotherapy treatments compared to the ER-negative subtype. The MYC v1 and v2 scores, specifically in ER-positive/HER2-negative tumors, were found to correlate to tumor aggressiveness. We cannot help but speculate that in the future the MYC score could be used as a prognostic biomarker to identify patients who will have increased tumor aggressiveness and worse prognosis. This will be particularly useful for potential patient selection for aggressive therapy, especially if the *MYC* oncogene can be used as a target for therapy.

Even though these data are novel, our study has a few limitations. First, although we examined three large patient cohorts and validated our findings, this remains a retrospective study. Future prospective studies will be required to further establish the clinical significance of the MYC targets v1 and v2 scores in ER-positive tumors. Additionally, our data are limited to only thousands of patients in total and mainly represent populations in North America and Europe. More study will be needed to evaluate the use of the MYC targets score in diverse populations. Further limiting this study is the fact that the MYC targets, although well studied, comprise a gene set that may be reduced in the future to a few genes that represent the greatest of the effects seen in the MYC targets set as a whole. This will require further study and may be applicable to a broader subset of patients should results differ. Lastly, our score is dependent on gene expression data, which is not currently included in the standard workup of breast cancer, and will therefore limit current clinical use of the MYC targets score.

In conclusion, this study has shown the value of MYC targets scores in predicting tumor aggressiveness in ER-positive/HER2-negative breast tumors. This supports further investigation into MYC scores and their association with treatment response in order to further utilize these scores in patient treatment.

## 4. Materials and Methods

### 4.1. The Cancer Genome Atlas Breast Cancer Cohort

We used the Pan Cancer Clinical Data Resource [15] and the cBio Cancer Genomic Portal [34] to obtain data regarding tumor gene expression and clinical data from the TCGA-BRCA project, which our group has previously reported [35,36]. There were 1065 tumors that were studied in total. Due to the TCGA data not including pathologic grade, Nottingham histologic grade was used instead which had previously been extracted from pathology reports from 573 of 1065 patients using Text Information Extraction System Cancer Research Network [37].

### 4.2. Data of METABRIC and Other Cohorts

Information regarding gene expression and correlating clinical data was collected from the METABRIC cohort (n = 1904) using the cBio portal system which our group has previously described [38,39,40]. We used the data from Sinn et al. (GSE124647, *n* = 140) to study the association of MYC score in metastatic tumors [17].

### 4.3. Gene Set Expression Analysis

We used log_2_-transformed normalized gene expression data. The two gene sets for which information was obtained were the “HALLMARK_MYC_TARGETS_V1” and the “HALLMARK_MYC_TARGETS_V2” gene sets of the Molecular Signatures Database Hallmark gene set collection [18]. We used gene set variation analysis (GSVA) to obtain gene expression data for these sets. Within both cohorts, studied tumor samples were placed into high and low MYC groups with upper one-third GSVA as the cut off. We used gene set enrichment analysis (GSEA) software and hallmark gene set collection for pathway analysis, as we have previously described [11,30,41,42,43,44,45]. We used a false discovery rate of 0.25, as previously recommended for GSEA.

### 4.4. Other

Features of tumors, including the composition of infiltrating immune cells, were estimated from gene expression data using xCell algorithm [46]. Statistical analyses and data plotting were performed using R software and Microsoft Excel. A *p*-value threshold of 0.05 was used to declare statistical significance. To determine significance among different groups we used one-way ANOVA or Fisher’s exact, as we described in legends. For survival analysis, the Kaplan–Meier method with log-rank test was used. The correlation between individual gene expression and MYC score was also obtained to evaluate for positive or negative relation to MYC score. For Appendix A, proliferation correlation was used to obtain the R value and COX analysis was used to obtain the hazard ratio (HR).

## Figures and Tables

**Figure 1 ijms-21-08127-f001:**
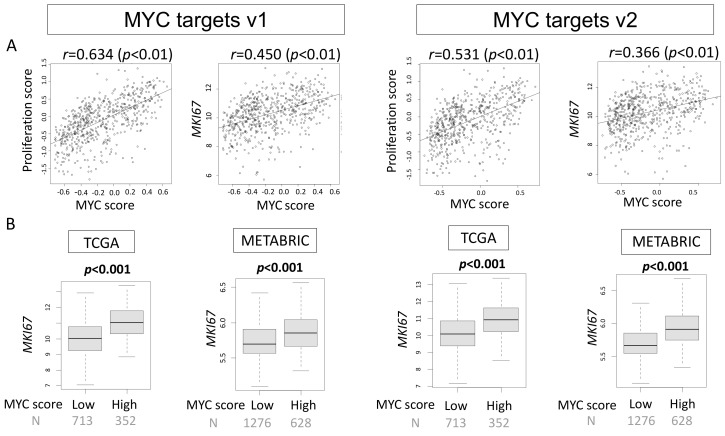
Association of the MYC targets v1 and MYC targets v2 scores with the proliferation score and expression of the *MKI67* gene (encodes KI67, the most commonly used marker for cell proliferation) in the TCGA and METABRIC cohorts. (**A**) MYC targets v1 and MYC targets v2 correlate with the proliferation score and expression of *MKI67* in the TCGA cohort. (**B**) High MYC targets v1 and MYC targets v2 score groups are significantly associated with increased expression of *MKI67* in both TCGA and METABRIC cohorts. Spearman rank correlation and one-way ANOVA were used for the analysis.

**Figure 2 ijms-21-08127-f002:**
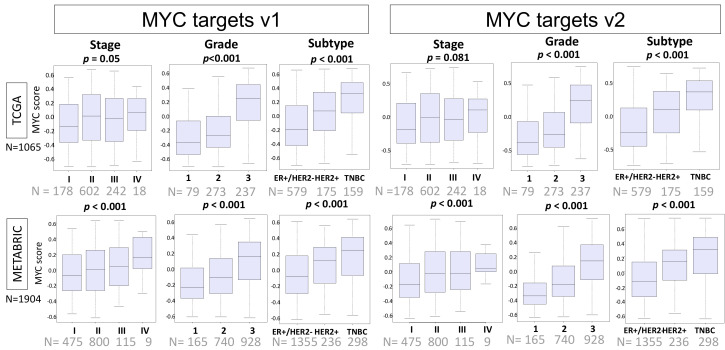
Association of the MYC score with clinical and pathologic features. Boxplots showing high and low MYC v1 and v2 scores in TCGA and METABRIC cohorts compared in American Joint Committee on Cancer (AJCC) cancer stage, Nottingham pathologic grade, and breast cancer subtypes. One-way ANOVA was used for the analysis.

**Figure 3 ijms-21-08127-f003:**
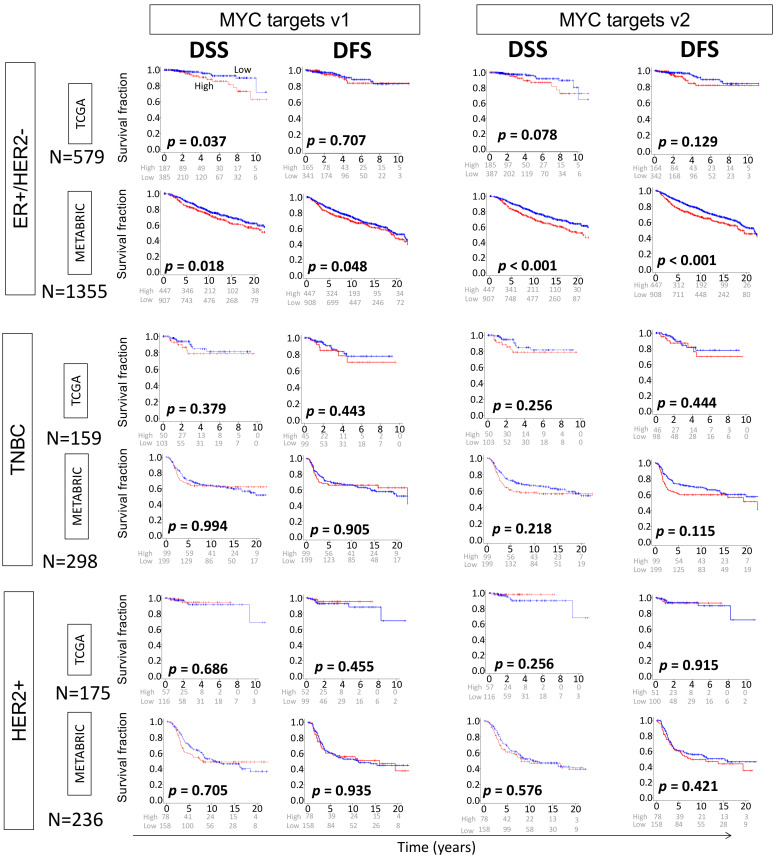
Association of the MYC score and clinical outcomes in the TCGA and METABRIC cohorts. Kaplan–Meier plots with log-rank test *p*-values showing survival difference among high and low MYC target v1 and MYC target v2 scores in the ER-positive/HER2-negative, TNBC and HER2-positive subtypes. DFS, disease-free survival; DSS, disease-specific survival.

**Figure 4 ijms-21-08127-f004:**
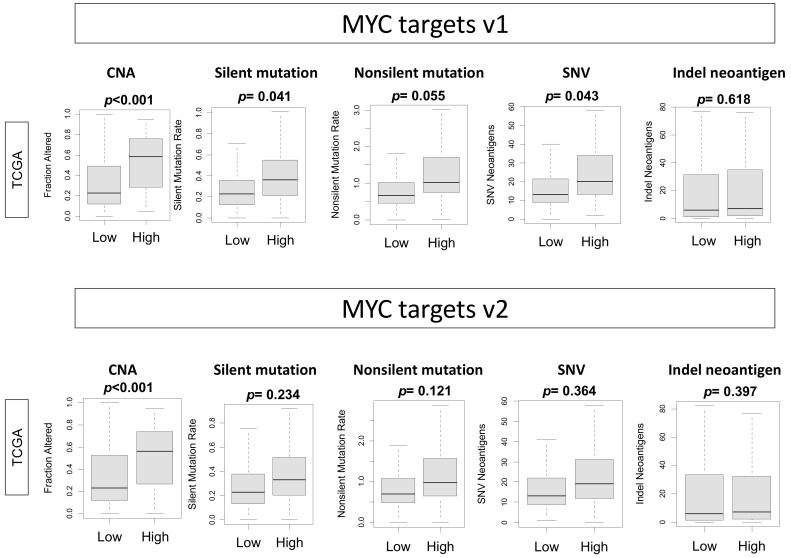
Association of the MYC targets v1 and v2 scores with mutation-related scores in ER-positive/HER2-negative breast cancer in the TCGA cohort. Boxplots depicting high and low MYC v1 and v2 scores associated with mutations assessed as copy number alteration (CNA), silent and nonsilent mutations, single-nucleotide variation (SNV), and insertion and deletion (Indel) mutations. One-way ANOVA was used for analysis.

**Figure 5 ijms-21-08127-f005:**
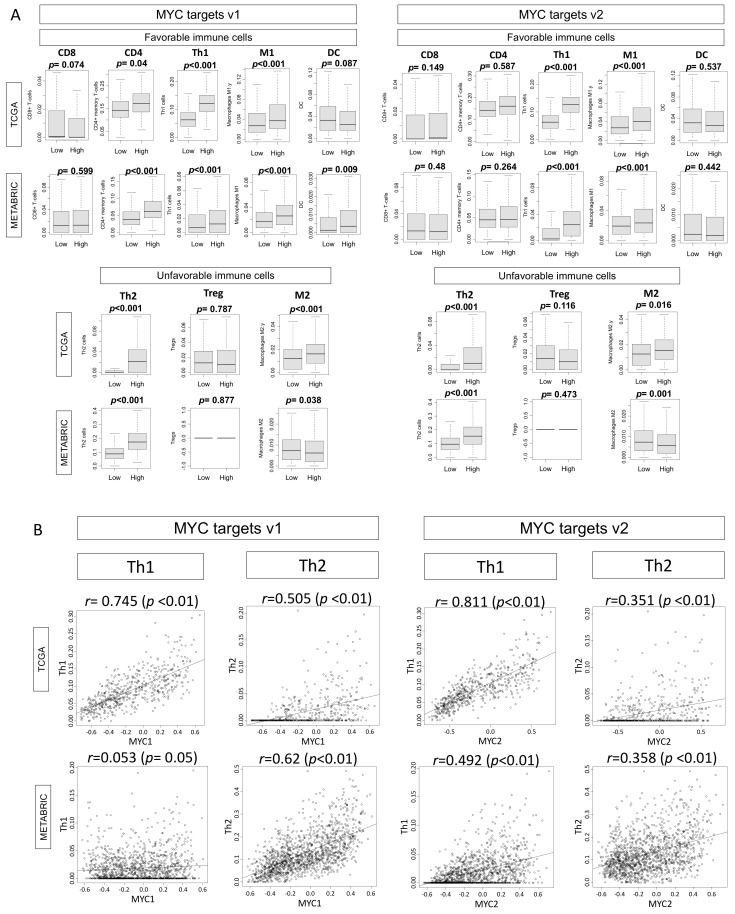
Association of the MYC targets v1 and v2 scores with fraction of tumor infiltrating immune cells in ER-positive/HER2-negative breast cancer. (**A**) Boxplots of high and low MYC v1 and v2 scores and infiltration of immune cells. Favorable immune cells: CD8, CD4, T helper type I (Th1) cells, M1 macrophages, dendritic cells (DC); unfavorable immune cells; T helper type 2 (Th2) cells, regulatory T (Treg) cells, M2 macrophages in ER-positive/HER2-negative breast cancer in TCGA and METABRIC cohorts. (**B**) Correlation of the MYC targets v1 and v2 scores with fraction of Th1 and Th2 cells in ER-positive/HER2-negative breast cancer in the TCGA and METABRIC cohorts.

**Figure 6 ijms-21-08127-f006:**
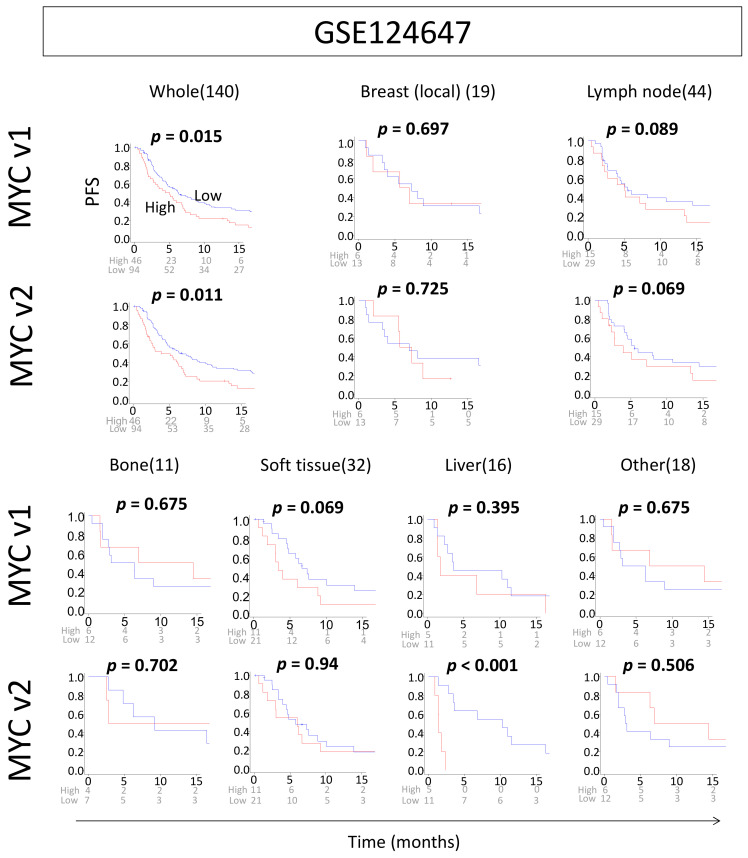
Progression-free survival (PFS) in metastatic breast cancer in high and low MYC targets v1 and v2 scores. Kaplan–Meier plots with log-rank test *p*-values showing PFS difference between high- and low-MYC score groups in different locations of metastatic disease in the GSE124647 cohort. Number of patients are written next to the organ in brackets.

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
