# Peer review of "MYC Targets Scores Are Associated with Cancer Aggressiveness and Poor Survival in ER-Positive Primary and Metastatic Breast Cancer"

_ijms, 2020, doi:10.3390/ijms21218127_

Round 1

Reviewer 1 Report

The manuscript submitted by Schulze A. et al presented the correlations of MYC targets v1 and MYC targets v2 scores with aggressiveness and survival in primary and metastatic breast cancer. The article is well organized and written; it contains impressive statistics but needs some improvements before it can be processed further. Here are my comments.

Introduction

  • the authors should present the other multigene risk score for breast cancer, the characteristics, and limitations

Results

A patient flow chart from all three cohorts would be useful.

Fig 1 A - the authors should delete TCGA close to the proliferation score

For a better understanding of the results, the authors can include the N for TCGA and METABRIC cohorts in fig. 1, 2 - similar to figure 3. 

Discussions

The discussions section needs improvements, the authors discuss their results, but the comparison to other tests used in breast cancer is scarce, l. 215 - 219.

The authors should present all the limitations of this study.

Materials and methods.

Should omit references 30, 31, 33, and 40.

Please check all references carefully; the authors missed a number of the article or pages.

Reviewer 2 Report

In this computationally-based analysis, Schulze et al examined the transcriptomes of 3108 breast cancers from three large data bases and correlated several biological features and patient survival with the expression of two sets of Myc target genes. High scores were associated with higher rates of tumor cell proliferation, shorter survival. In ER+ cases, the V1 Myc target gene set was associated with tumors having the highest mutational burdens. Both the V1 and V2 subsets correlated with increased infiltration. High Myc scores were also associated metastatic tumors.

Major comments:

1. The difference between V1 and V2 subsets of Myc target genes should be explained rather than just referring to a reference. Are they all direct targets? If so, how were they identified? By ChiP, or by other means? etc. What is the rationale for keeping them as separate groups? This would best be done in the Results section when the two groups are first introduced. It might also help put into context why one set is more predictive certain than another.

2. Another factor that might help if for the authors to include in Suppl. Table whether the target genes are positive or negative Myc targets. My guess is that the V1 and V2 sets contain both types of targets. It’s not entirely clear from the analyses performed whether the correlations shown in Fig. 1 took this into account. If not, then the positive and negative targets might tend to cancel one another out, thereby leading to correlation coefficients that are not as robust as they could be if the positive and negative targets were considered separately

3. The distinction between V1 and V2 targets is made further confusing by the fact that several genes are included in both sets (MYC itself, CDK4, DDX18, etc.)

4. Are there any V1 and/or V2 targets that, either individually or as a group, are particularly good at identifying highly aggressive tumors and/or better or worse survival? Perhaps the entire set can be further boiled down to a critical few.

5. Related to (4), one of the major conclusions of the study as stated in the Abstract and lines 126-127 is that “This suggests that MYC target scores are clinically relevant specifically in the ER-positive/HER2-negative subtype.” Perhaps a more refined subset of target genes based on the above-discussed criteria would be more sensitive at distinguishing biological/survival differences among these other breast cancer groups.

6. In Fig. 4, the authors show that mutational burden correlates with Myc target gene dysregulation. Myc over-expression has a well-known tendency to directly impact mutation and DNA copy number, in part due to its ability to induce ROS and to promote aberrant S-phase progression in the absence of mitosis that may be dependent on p53 status (Mol Cell Biol. 1999,19:5339-51. Cancer Res. 2001,61:6487-93, Mol Cell. 2002,9:1031-44.). It is with this data set where the correlation with Myc levels rather than the entire Myc target gene data set might better correlate with the mutational burden.

.Minor comments:

In Fig. 1, the authors show fairly robust correlations between V1 and V2 target genes and preoliferation. However, they do not mention whether Myc itself (which is one of the members of the V1 set) gives the same correlative results

Round 2

Reviewer 2 Report

No comments.  My concerns have been well-addressed